

# Impact of synthetic spaceborne NO₂ observations from the Sentinel-4 and Sentinel-5p platforms on tropospheric NO₂ analyses

Renske Timmermans[1], Arjo Segers[1], Lyana Curier[2], Rachid Abida[3], Jean-Luc Attié[3,4], Laaziz El Amraoui[3], Henk Eskes[5], Johan de Haan[5], Jukka Kujanpää[6], William Lahoz[7], Albert Oude Nijhuis[8], Samuel Quesada[3], Philippe Ricaud[3], Pepijn Veefkind[5], and Martijn Schaap[1,9]

[1] TNO, Netherlands Organisation for Applied Research, Climate, Air and sustainability division, 3508 TA Utrecht, The Netherlands
[2] TNO, now at CBS, Central Bureau of Statistics, Heerlen, The Netherlands
[3] CNRM-GAME, Météo-France/CNRS UMR 3589, Toulouse, France
[4] Université de Toulouse, Laboratoire d'Aérologie, CNRS UMR 5560, Toulouse, France
[5] Royal Netherlands Meteorological Institute (KNMI), P.O. Box 201, 3730 AE De Bilt, the Netherlands
[6] Finnish Meteorological Institute (FMI), P.O. Box 503, 00101 Helsinki, Finland
[7] NILU – Norwegian Institute for Air Research, P.O. Box 100, 2027 Kjeller, Norway
[8] KNMI, now at SkyEcho, Rotterdam, the Netherlands
[9] FUB – Free University Berlin, Institut für Meteorologie, Carl-Heinrich-Becker-Weg 6-10, 12165 Berlin, Germany

*Correspondence to*: Renske Timmermans (renske.timmermans@tno.nl)

**Abstract.** We present an Observing System Simulation Experiment (OSSE) dedicated to the evaluation of the added value of the Sentinel 4 and Sentinel 5P missions for tropospheric nitrogen dioxide (NO₂). Sentinel 4 is a geostationary (GEO) mission covering the European continent, providing observations with high temporal resolution (hourly). Sentinel 5P is a low-Earth Orbiting (LEO) mission providing daily observations but with a global coverage. The OSSE experiment has been carefully designed, with separate models for the simulation of observations and for the assimilation experiments, and with conservative estimates of the total observation uncertainties. In the experiment we simulate Sentinel 4 and Sentinel 5P tropospheric NO₂ columns and surface ozone concentrations at 7 by 7 km resolution over Europe for two three-month summer and winter periods . The synthetic observations are based on a nature run (NR) from a chemistry transport model (MOCAGE) and error estimates using instrument characteristics. We assimilate the simulated observations into a chemistry transport model (LOTOS-EUROS) independent from the NR to evaluate their impact on modelled NO₂ tropospheric columns and surface concentrations. The results are compared to an operational system where only ground-based ozone observations are ingested. Both instruments have an added value on analysed NO₂ columns and surface values, reflected in decreased biases, and improved correlations. The Sentinel 4 NO₂ observations with hourly temporal resolution benefit modelled NO₂ analyses throughout the entire day where the daily Sentinel 5P NO₂ observations have a slightly lower impact that lasts up to 3-6 hours after overpass. The evaluated benefits may be even higher in reality as the applied error estimates were shown to be higher than actual errors in the now operational Sentinel 5P NO₂ products. We show that an accurate representation of the NO₂ profile is crucial for the benefit of the column observations on surface values. The results support



the need for having a combination of GEO and LEO missions for NO₂ analyses in view of the complementary benefits of hourly temporal resolution (GEO, Sentinel 4) and global coverage (LEO, Sentinel 5P).

## 1    Introduction

Air pollution (indoor and outdoor) is responsible for one out of nine deaths worldwide (WHO, 2016) and is one of the biggest environmental threats for our living planet. Outdoor air pollution alone causes about 3 million premature deaths per year (WHO, 2016). The main pollutant accountable for this significant health impact is particulate matter (PM or aerosols), consisting of small particles in the atmosphere that enter the lungs and blood stream and cause cardiovascular, cerebrovascular and respiratory impacts. The origin of PM is direct emissions of small particles, or formation in the atmosphere via chemical reactions involving species emitted as gases. Nitrogen dioxide ($NO_2$) is one of the main precursors for this secondary formation of particulate matter, as it is a source for the formation of nitrate aerosols (Seinfeld and Pandis, 2006). At high loadings, $NO_2$ by itself is also toxic, and long-term exposure to elevated levels of $NO_2$ such as currently observed in cities throughout the world has also been linked to reduced lung function growth (WHO, 2018). The main sources of emissions of $NO_2$ are combustion processes (traffic engines, heating and power generation). To allow the formulation of effective policy measures for reducing the exposure to air pollution, accurate knowledge on the sources and distribution of air pollutants is required. This knowledge is gained through observations of the atmospheric composition by ground-based and satellite instruments. To obtain a full picture in both space and time, these observations are combined with models that take into account all relevant processes in the atmosphere influencing the distribution of pollutants, forming the basis for data assimilation (Bocquet et al., 2015). The synergetic use of models with observations provides the best possible estimate of the 3-dimensional distribution of air pollutants in the atmosphere in the past ((re)analyses), current (nowcasts) and the future (forecasts).

$NO_2$ is one of the atmospheric components with the longest observation record from space (Boersma et al., 2018; Hilboll et al., 2013). Due to the large concentrations in the boundary layer, a strong $NO_2$ signal can be observed from space despite the reduced sensitivity of satellite instruments to boundary layer concentrations owing to molecular scattering in the UV. The Global Ozone Monitoring Instrument (GOME) was one of the first satellite instruments to provide a long timeseries of tropospheric $NO_2$ columns (Burrows et al., 1999). The standard spatial resolution of these GOME observations was 40 by 320 km. Since then, there has been a development of newer instruments with increasing spatial resolution: the SCanning Imaging Absorption SpectroMeter for Atmospheric CHartographY (SCIAMACHY) (Bovensmann et al., 1999) with 30 x 60 km, GOME-2 with 40 x 40 km and the Ozone Monitoring Instrument (OMI) (Levelt et al., 2006) with 13 x 24 km resolution. Each instrument provides a more detailed view of $NO_2$ concentrations near the surface with their higher spatial variability. These observations have been successfully used to improve air quality analyses (Inness et al., 2018; Silver et al., 2013; Wang et al., 2011); to derive NOx emissions at different scales (e.g. Beirle et al., 2011; de Foy et al., 2015; Ding et al., 2017; Mijling et al., 2013; Zhang et al., 2018); and to estimate trends in concentrations and emissions (e.g. Castellanos and





Boersma, 2012; Curier et al., 2014; de Ruyter de Wildt et al., 2012; Hilboll et al., 2013; Konovalov et al., 2010; Lamsal et al., 2015; Liu et al., 2017; Lu et al., 2015; Paraschiv et al., 2017; Schneider et al., 2015; Zhou et al., 2012). However as $NO_2$ has a short lifetime ranging from a few hours in summer to a day in winter (Seinfeld and Pandis, 2006), its concentration is highly variable in space and time, consequently exceedances of limit values are usually very locally dependent. There is, therefore, a need for improved information of pollutant concentrations at higher spatial (urban scales or even street level) and temporal (hourly) resolution.

Copernicus is the current European Union programme for the establishment of a European capability for Earth Observation (http://www.copernicus.eu). It includes a set of services such as the Copernicus Atmosphere Monitoring Service CAMS (https://atmosphere.copernicus.eu/)**,** aimed at providing consistent and quality-controlled information related to air pollution and health, solar energy, greenhouse gases and climate forcing, across the globe. The CAMS service encompasses a global and regional air quality forecast and analyses service. At the base of the Copernicus services lie data from satellite Earth Observation systems and in situ (non-space) networks. The Copernicus programme also encompasses a space component dedicated to new spaceborne missions developed and managed by the European Space Agency (ESA) and the European Organisation for the Exploitation of Meteorological Satellites (EUMETSAT). Three of these new missions will be delivering atmospheric composition products, including tropospheric $NO_2$ columns at an unprecedented spatial resolution of 3.5 to 7 km and with improved signal to noise ratio as compared to predecessors. The launch of the first of these three missions, Sentinel-5 Precursor (S5P) (Veefkind et al., 2012) on board of a low-earth orbit (LEO) platform, took place in October 2017. S5P is the successor of the OMI mission and will be followed up by the Sentinel-5 (S5) mission planned for launch in 2021. On board the S5P mission, the TROPospheric Ozone Monitoring Instrument (TROPOMI) provides tropospheric $NO_2$ at a resolution of 7 by 3.5 km, as compared to 13 by 24 km for OMI. First results show the large potential of the instrument for providing insight on the distribution of $NO_2$ at high-resolution, resolving individual larger industrial complexes and cities and the resulting plumes on a daily basis (www.tropomi.eu). The satellite flies in an early afternoon sun-synchronous orbit with an equator crossing mean local solar time of 13:30 with a wide swath enabling daily global coverage but limiting the temporal coverage to one or two daytime observations per day at mid-latitudes. The future S5 mission (ESA, 2018b) is expected to provide similar, or somewhat lower, resolution than TROPOMI. S5 will fly in an orbit with a morning equator crossing mean local solar time of 9:30, and will follow-up and complement the S5P data. The Ultra-violet/Visible/Near-Infrared (UVN) sounder as part of the Sentinel-4 (S4) mission on board the geostationary (GEO) Meteosat Third Generation-Sounder (MTG-S) satellite (ESA, 2018a), with a planned launch in 2021, will also provide similar resolution to TROPOMI, but with higher temporal resolution (hourly). These hourly observations will allow the monitoring of the $NO_2$ diurnal cycle over Europe.

While the planning of new satellite missions and development of dedicated instruments is a long and costly endeavor, Observing System Simulation Experiments (OSSEs) are designed to allow objective determination of the added value and impact in comparison to current operational observing systems (Lahoz and Schneider, 2014) and to assess the value of different instrument observing designs. OSSEs are extensively used in meteorological practices for determining the added



value of new observing systems for weather forecasts (e.g. Atlas, 1997; Atlas et al., 2003) and in the past 10 years have evolved for air quality applications (Timmermans et al., 2015). These OSSEs have focused on aerosols (Descheemaecker et al., 2018; Timmermans et al., 2009; Yumimoto and Takemura, 2013), carbon monoxide (CO) (Abida et al., 2017; Claeyman et al., 2011; Edwards et al., 2009; Yumimoto, 2013) and ozone (Hamer et al., 2011; Zoogman et al., 2011, 2014a, 2014b)

from either LEO, GEO or a combination of both observing systems.. The review by Timmermans et al., (2015) provides a framework and set of requirements for each step in the framework to ensure realistic evaluation of the benefit of the new instruments. The main requirements relate to the representation of the real atmospheric situation and the simulation of realistic observations and their associated errors. Currently, the potential benefits of the planned S4 on top of those from S5P have not been quantified.

In this paper, we describe an OSSE dedicated to the quantification of the impact of the S5P and S4 observations of $NO_2$ for improving European air quality surface analyses. We investigate the benefits of both instruments separately but also combined. At the time of the study, S5P was not launched yet, requiring the application of an OSSE instead of an OSE (Observing System Experiment), used to assess the added value of existing observations. We follow the approach and requirements described in Timmermans et al. (2015) to ensure the robustness of our results and avoid overoptimistic results.

This work was part of a study funded by ESA called "Impact of Spaceborne Observations on Tropospheric Composition Analysis and Forecast" (ISOTROP), to study the impact of S4, S5 and S5P observations of ozone (Quesada-Ruiz et al., 2019), CO (Abida et al., 2017), $NO_2$ and HCHO on air quality analyses.

The structure of the paper is as follows. In Sect. 2, we describe the different components of the OSSE. Sect. 3 provides the results, including first an evaluation of our representation of the true situation and second an evaluation of the added value of

the S5P and S4 $NO_2$ observations for air quality analyses. Finally, Sect. 4 presents the conclusions, discussion of the results and identification of further work.

## 2    The OSSE set-up

In this study, we follow the different OSSE steps and requirements as identified in Timmermans et al. (2015). Figure 1 provides a schematic overview of the work flow in this study. As the observations under investigation were not available yet,

we first had to produce a set of synthetic observations. We based production of the simulated observations on a so-called nature run (NR), which acts as the representation of reality. We converted the output from the NR into synthetic observations using information on the S4 and S5P instrument characteristics, as well as influencing parameters for the observations such as clouds and surface albedo. Next, the synthetic observations were ingested into the chemistry transport model (LOTOS-EUROS) using an ensemble Kalman Filter data assimilation system. The resulting modelled atmospheric composition fields

(the assimilation runs, AR) are then compared to the NR, a control run (CR) without assimilating any observations and a reference run (RR) assimilating current operational observations, to evaluate the benefit of the synthetic observations. When comparing to a RR, the benefit is evaluated in comparison to current operational system capabilities. One of the main




requirements for a realistic OSSE is the preferred use of different models for the NR and the CR/AR. Using the same model can lead to the identical twin problem (also referred to as inverse crime in the mathematical inverse problem literature (Kaipio and Somersalo, 2005)) and overoptimistic OSSE results (Arnold and Dey, 1986). In this OSSE, the MOCAGE (MOdèle de Chimie Atmosphérique de Grande Echelle) model (Peuch et al., 1999) provides the NR, while the LOTOS-

EUROS model (Manders et al., 2017) provides the CR and AR. To further avoid the identical twin problem and introduce differences between the model results, the model systems were forced using different meteorological drivers and emission information.  In the next sections, we provide more details on the individual OSSE components including these two models.

We have set-up the study for two three-month study periods. The first three month period (June to August 2003) includes the 2003 heat wave over Europe. The stagnating weather conditions with reduced horizontal transport of the air masses and very

warm temperatures lasting several days led to highly elevated levels of ozone and CO. At the same time, the chosen timeframe also covers normal conditions, allowing us to look at the full range of pollution levels occurring in a summer season in Europe. Additionally, we chose a three month winter period (November 2003 – January 2004) to cover different seasons and chemical situations.

**2.1    Nature run**

The objective of the NR is to represent the true state of the atmosphere, forming the basis for the simulation of observations. The main requirements for the NR in an air quality OSSE following Timmermans et al., (2015) are that it is produced using a high performance state-of-the-art air quality model significantly different from the model used for the assimilation runs. The NR concentrations should show spatial and temporal variations in accordance to real representative observations, cover

different seasons and an extended geographical region. The resolution should be sufficient to resolve the variability at the scale of the observations of interest.

The NR in this OSSE is performed using the MOCAGE model (Peuch et al., 1999), a chemistry transport model developed at Météo-France. The model is operationally applied at Météo-France to provide the national chemical weather forecasts (Dufour et al., 2005) and is part of the regional Copernicus Atmosphere Monitoring Service (CAMS) ensemble which

provides operational air quality forecasts and analyses on a daily basis over Europe (http://macc-raq.copernicus-atmosphere.eu/).  In this context, the model calculations are regularly evaluated against both observations and results from the other models in the ensemble (Marécal et al., 2015).

The NR is constructed using a two-way nested configuration: a European grid (15W-35E, 35N-70N) with a 0.2º x 0.2º horizontal resolution and a smaller regional grid (5W-10E,41N-53N), covering France and surrounding regions, with a 0.1º x

0.1º horizontal resolution. The MOCAGE model includes 47 sigma-hybrid vertical levels from the surface up to 5 hPa. The vertical resolution is 40 to 400 m in the boundary layer (seven levels) and approximately 800 m near the tropopause and in the lower stratosphere. The anthropogenic emissions are based on the TNO-MACC I  inventory (Kuenen et al., 2011), complemented by EMEP 0.5º x 0.5º shipping emissions. Biogenic emissions are fixed monthly using the Simpson approach



(Simpson et al., 1995). Dynamically, the model is forced every 3 hours by meteorological data from the Météo-France analysis data of the ARPEGE model (Courtier et al., 1991).

## 2.2 Observation simulator

We generate $NO_2$ tropospheric column synthetic observations using model profiles extracted from the NR model datasets. The observations are generated for the TROPOMI instrument on board the S5P satellite and the UVN instrument on board the S4 satellite. The TROPOMI instrument is a spectrometer based on its predecessors OMI and SCIAMACHY (Veefkind et al., 2012). It measures in the ultraviolet (UV) – visible (270–500 nm), near infrared (NIR, 675–775 nm) and short-wave infrared (SWIR, 2305–2385 nm) wavelengths ranges to enable the retrieval of several air quality data products, including ozone, $NO_2$, formaldehyde, $SO_2$, methane, and CO. The instrument has a wide swath of 2600 km allowing daily global coverage and was designed for a spatial resolution of 7 by 7 km. After our study and the launch of the instrument, the spatial resolution was further improved to 7 by 3.5 km. The S4 / UVN instrument (ESA, 2018a) is a spectrometer measuring in the UV (305-400 nm), visible (400-500 nm) and NIR (750-775 nm) wavelength ranges. Over Europe, it will have a spatial resolution of 8 km and an hourly temporal resolution.

The generation of the synthetic observations involves the following steps:

i)   The generation of the S4 and S5P orbits, geolocations of the individual high-resolution observations and their corresponding geometrical properties (solar, viewing and azimuth angles). We do this for the appropriate overpass time (S5P) or observation time (S4).

ii)  Based on the cloud distribution of the European Centre for Medium-Range Weather Forecasts (ECMWF) weather model analyses, we simulate effective cloud fractions as would be observed by the satellite.

iii) We generate lookup tables to compute the scene-dependent averaging kernels and observation uncertainties.

iv)  We interpolate the NR fields in space and time to the observation footprints to derive a set of synthetic observations for the three summer months and the three winter months.

We discuss in more detail below the individual steps performed for the TROPOMI instrument on board the S5P satellite and the UVN instrument on board the S4 satellite.

### 2.2.1 Orbit simulator

We simulate the geometry of the S5P and S4 orbits and field of view using the System Tool Kit (STK, developed by AGI, http://www.agi.com/products/). Using the sentinel orbit characteristics, the STK provides the time-dependent geolocation of the edges of the swath. Based on the location of the edges, we compute the coordinates of the individual observations assuming a spatial resolution of 7x7 $km^2$ at nadir for both instruments. We first compute the geometry of one S5P orbit (S4 has field of view over Europe), and then apply appropriate time and longitude shifts to obtain all the orbits needed for both study periods. Subsequently, we compute the geometry for each individual observation (the solar, viewing and azimuth





angles). Note that normally the size of the footprints away from nadir increases roughly like 1/viewing angle. For S5P the size of the real footprints at the edges of the swath are reduced by a factor 2 by changing the binning, leading to a more uniform footprint size across-track. The footprints of the generated synthetic observations are 7km at nadir up to ~8.5km at the edges. This is not fully realistic but approximating the real footprints when using the binning.

As a final step, we store only the part of the orbit that has an overlap with the model domain. The data are stored in formatted files that mimic the format of current NO$_2$ column products from satellite retrievals.

### 2.2.2   Cloud, temperature and surface albedo information

The satellite measurements of S4 and S5P, once available, will be used to retrieve cloud properties like cloud fraction and cloud pressure, using either the O$_2$-A band (Wang et al., 2008) or the O$_2$-O$_2$ absorption feature around 477nm (Veefkind et

al., 2016). However, for an OSSE those need to be estimated differently. For this purpose we take cloud information and other relevant model fields from the ECMWF weather forecast archive for the two three-month study periods at a resolution of 0.25 degrees. The retrieved fields are temperature, pressure, liquid and ice water content, specific humidity and cloud fraction. To simulate the cloud parameter observations, we convert the ECMWF cloud quantities to cloud optical properties, which determine the reflectance at the top of the atmosphere. Based on these reflectances, we simulate the effective cloud

fraction and effective cloud top height similar to the procedure in the O$_2$-O$_2$ cloud retrieval. The distribution of effective cloud fractions obtained in this way was compared with OMI O$_2$-O$_2$ cloud observations for the year 2006 and a reasonable qualitative agreement of the histograms was found for summer and winter months, with somewhat less cloud free days and more intermediate cloud fractions in the OMI dataset, which could in part be due to the above-average number of sunny days in 2003 (Williams et al., 2013). We use the simulated effective cloud fraction and cloud pressure in the synthetic retrievals.

We convert the cloud fraction into a cloud radiance fraction (the fraction of the top-of-atmosphere radiance coming from the cloud-covered part of the scene) by computing radiances using the surface albedo map and assuming a cloud albedo of 0.8. The surface albedo is taken from the 5-year OMI Lambertian reflectivity dataset, extended from the 3-year dataset published by Kleipool et al. (2008). Figure 2 shows an example of the cloud radiance fraction simulated in this way for the13:30 local time afternoon overpass of S5P on a day with mainly cloud-free conditions over continental Europe.

### 2.2.3   Averaging kernel lookup tables

The ideal approach for the generation of synthetic observations would involve the following steps. (1) Use the NR profile and radiative transfer model to simulate radiances at the top of the atmosphere. (2) Add noise to the simulated radiances based on the satellite specifications. (3) Apply the retrieval approach to the simulated radiances, and add extra noise related to uncertainties in retrieval parameters such as cloud fraction and surface albedo. Due to the large number of observations

provided by the sentinels, this approach is not feasible.

Following Rodgers (2000), we note that applying the forward radiative transfer model, followed by the retrieval is equivalent (after linearization) to applying the kernel matrix **A**, and the synthetic observations $x_r$ can be generated with the equation $x_r =$





$x_a + \mathbf{A}\,(\,\mathbf{x} - \mathbf{x_a}\,) + \epsilon$, where $x_a$ is the a-priori column/profile and $\epsilon$ is the error due to the instrument and retrieval errors. As shown by Eskes and Boersma (2003), for differential optical absorption spectroscopy (DOAS) column retrievals $(\mathbf{A} - \mathbf{I})\,x_a = 0$, and the equation simplifies to $x_r = \mathbf{A}\,\mathbf{x} + \epsilon$. Here $\mathbf{x}$ is the true vertical profile of $NO_2$ and the averaging kernel is a 1D vector. For the DOAS column retrievals considered here, we compute the elements of the averaging kernel from the height-

dependent air-mass factors (AMF), or box AMFs (Eskes and Boersma, 2003).

Using the radiative transfer toolbox DISAMAR (Determining Instrument Specifications and Analyzing Methods for Atmospheric Retrieval) (de Haan, 2012), we generate lookup tables for the box AMFs for the range of geometries of the S4 and S5P. Results are stored for 21 pressure levels between 1050 and 0.1 hPa, 9 surface albedos, 10 cloud/surface pressures, 10 solar zenith angles, 15 viewing zenith angles, and 3 relative azimuth angles. We use linear interpolation to find the AMF

values for the satellite geometries.

### 2.2.4    Observation uncertainties

The Sentinel 4-5 Mission Requirements Document (ESA, 2011) specifies signal-to-noise ratios for the Sentinels 4 and 5(P). With the DISAMAR tool, we can translate these irradiance/radiance requirements to $NO_2$ retrieval uncertainties using either DOAS or optimal estimation. We experimented with this, but found that the uncertainty estimates are very sensitive to

assumptions on spectral correlations in the noise. For the final set of simulations, we adopted the DOAS approach following an early version of the TROPOMI $NO_2$ Algorithm Theoretical Baseline Document (ATBD) (van Geffen et al., 2014). We base this approach on our experience with the OMI instrument. We extrapolate the uncertainty estimate for TROPOMI from the OMI DOAS slant column retrievals, and obtain an absolute estimate of $0.7\ 10^{15}$ molecules $cm^{-2}$ for $NO_2$ (of the order of 10% for moderately polluted conditions) for the slant column, nearly independent of geometry/latitude. We use this estimate

in the retrievals.

Other uncertainties due to retrieval input parameters, affecting the AMFs, are propagated through the DOAS equations following the approach described in Boersma et al. (2004). We assume fixed uncertainties for the cloud pressure (50 hPa), the cloud fraction (0.02), and the surface albedo (0.015). We do not explicitly model the stratospheric NO2 distribution, but implicitly assume that a method will be adopted to estimate the stratospheric column, so that for the synthetic observations

only the uncertainty of the stratospheric estimate needs to be accounted for. Again, based on our experience with OMI, SCIAMACHY, GOME-2 (van Geffen et al., 2014), the stratospheric vertical column uncertainty resulting from this procedure is set to $0.15\ 10^{15}$ molecules $cm^{-2}$, or about 6% of an average mid-latitude stratospheric column.

### 2.2.5    Synthetic observations

We compute the synthetic observations by applying the averaging kernels to the values from the NR linearly interpolated to

the locations from the orbit simulator. We add gaussian noise to the retrievals making use of the estimated retrieval uncertainty (Sect. 2.2.4). Figure 3 gives an example of one local time 13:30 overpass of S5P, based on the NR of 1 June


2003. The bottom panel shows the MOCAGE NR tropospheric column amount. Due to the wind from the south, there is transport of pollution from the Benelux over the North Sea. We show the uncertainty modelled in the synthetic retrieval in the top right panel. We determine the detection limit noise level by combining the slant column noise with the stratospheric column uncertainty. The AMF retrieval errors related to cloud and albedo scale linearly with the retrieved column and are of
the order of 25-50%.

S4 will fly on a geostationary platform; this will provide additional diurnal information on $NO_2$. We simulate the S4 observations each hour during daytime. Figure 4 provides an example of the S4 observations as simulated, one in the early morning 4:00 UTC, and one during mid-day 12:00 UTC. The images demonstrate the large diurnal cycle present in the MOCAGE NR of $NO_2$.

**2.3   Synthetic ground observations**

To determine the added value of the new instruments, we evaluate the benefit of the new observations on top of the benefit from observations used in the current operational system. At the time of this study, only ozone observations from monitors at the surface were used in the LOTOS-EUROS operational forecasts. Ground-based $NO_2$ observations were not included as they measure $NO_2$, being affected by contamination from other oxidised nitrogen species (Peroxyacetyl Nitrate (PAN) and
Nitric acid ($HNO_3$)) (Giordano et al., 2015; Steinbacher et al., 2007). Ozone assimilation influences $NO_2$ through chemistry and through adjusted NOx emissions (see Sect. 2.5.1).

Synthetic ground-based ozone observations have been produced from the NR results to be consistent with the simulated satellite observations. We assume the network of currently existing stations for ozone is representative for the locations that will be available during the upcoming satellite missions. Therefore, the observation locations of the synthetic surface ozone
observations have been drawn from the existing AirBase network (version 6, Simoens (2012)). We use only stations classified as 'background', since these are representative for concentrations at the resolution of the models used in this study. Additionally, we omit stations above 700 m above sea level considering the focus of the project on boundary layer concentrations and the difficulties models have reproducing conditions in mountainous areas. Figure 5 shows the location of the selected stations. During data assimilation experiments, it is common practice to leave a number of observations out of
the assimilation procedure, and use these for validation. The split of observations used and presented in Figure 5 is taken from the MACC II project (Marécal et al., 2015). We use only the locations of stations labelled as assimilation stations. We sample the ground-based synthetic observations from the lowest level NR output by performing a bilinear horizontal interpolation to the selected station locations.

One of the most important quantities in an assimilation system is the representation of the variance of the departures, which
quantifies differences between an observation and its forecast by the model. These differences arise due to a combination of instrumental errors, the grid cell formulation used in a model, deficiencies in the simulated processes, and other errors. Proper quantification of the representation error, related to the mismatch of the point measurement and the model grid and effective resolution of the model, is a major task in an assimilation exercise. Generally, we base its estimate on comparison



of the observations and model simulations. The only exception is determination of the measurement error, which is solely dependent on the instruments. However, in air-quality databases like AirBase, it is not common practice to provide characteristics of instrumental errors (while retrieval uncertainties are provided for individual observations in the case of satellite measurements). Therefore, the representation error of ground-based observations should be set or parameterized

explicitly.

To ensure realistic synthetic surface ozone observations, we mask the synthetic ground-based observations if the corresponding real observations in AirBase are missing too. Secondly, we add a random number to each synthetic value to represent the instrumental error. We base the distribution of this random number on statistics obtained from the AirBase observations available for the same period. More specifically, we determine the temporal auto-covariance in a time series of

ozone observations. The difference gap from variance (lag zero) to the first co-variance (lag one) is then a measure of the random hour-to-hour variation. This variation therefore represents both instrumental errors as well as variations on timescales less than an hour.

### 2.4    Control run

The CR in the OSSE is performed using the LOTOS-EUROS model, a chemistry transport model developed by TNO

(Manders et al., 2017). The model provides the national Dutch operational chemical weather forecasts (https://www.lml.rivm.nl/verwachting/animatie.html), and is part of the regional CAMS ensemble, which provides operational air quality forecasts and analysis on a daily basis over Europe (http://macc-raq.copernicus-atmosphere.eu).  In this context, the model calculations are regularly evaluated against both observations and results from the other models in the ensemble (Marécal et al., 2015). The objective of the model is to describe air pollution in the lowermost atmosphere; to

achieve this, the standard version has four vertical layers following a dynamical mixing layer approach. The first layer is a fixed layer of 25-metre thickness; the second layer follows the mixing layer height. We evenly distribute the remaining two reservoir layers between the mixing layer height and 3.5 km. The implicit assumption of the LOTOS-EUROS model is the presence of a well-mixed boundary layer, i.e. constant concentrations between the surface layer and the mixing layer height.

The CR is constructed using a nested configuration: a European grid (Europe: 15W-35E, 35N-70N) with a 0.125º x 0.25º

resolution in the longitudinal-latitudinal direction and a smaller regional grid (Zoom: 5W-10E, 41N-53N) with a 0.0625º x 0.125º resolution in the longitudinal-latitudinal direction. The lateral and top boundary conditions for the European grid are taken from a global re-analysis by the TM5 model (Huijnen et al., 2010; Williams et al., 2017). The same TM5 model runs are also used for the extension of the LOTOS-EUROS runs from its top at 3.5 km to the tropopause.

We base the anthropogenic emissions on the TNO-MACC II inventory (Kuenen et al., 2014). Note this is a different

inventory version than used in the NR, supporting the requirement of significant differences between the models for the NR and the CR/AR.  Biogenic emissions are calculated online as described in Schaap et al. (2009). Fire emissions are taken from the MACC-II GFAS v1.0 product (Kaiser et al., 2012). These emissions vary from day to day, and are provided on a 0.5º resolution grid. Dynamically, we force the model by meteorological data from ECMWF. We choose this different



meteorological forcing from the one used in the NR to meet the requirements on the differences between the NR and the CR/AR.

## 2.5    Assimilation runs

To evaluate the benefit of the S4 (GEO) and S5P (LEO) NO$_2$ observations for air quality analyses, we assimilate the
synthetic observations into the LOTOS-EUROS model. We use different configurations to evaluate the individual and combined benefits of the GEO and LEO satellite instruments. We evaluate the benefit of the new observations against a reference situation where we assimilate only ground-based ozone observations.

### 2.5.1    Data assimilation system – Ensemble Kalman Filter

The LOTOS-EUROS set-up features an active data assimilation system based on the Ensemble Kalman Filter (EnKf)
technique (Evensen, 2003). A Kalman filter computes probability density functions (pdfs) of the true state, given 1) a transition model to propagate the state in time with associated uncertainties; and 2) observations with associated representation error. Starting from the initial pdf, the filter first performs a forecast step propagating the pdf in time to the first moment that observations become available. Then, during the analysis step, we replace the forecast pdf by an analysed version that takes into account the new information available. The Kalman filter is an example of sequential assimilation,
since forecast and analysis steps follow each other sequentially in time and use only information from the past. To be able to apply this technique to a large scale air quality model, the pdf  is described by an ensemble of model states in a so-called EnKF (Evensen, 2003). The spread between the ensemble members should describe the uncertainty in the state quantities as the mean and covariance of the state is computed from the ensemble statistics. The number of required ensemble members depends on the complexity of the pdf, commonly determined by the non-linearity of the transition model and the complexity
of the associated model uncertainty. In practice, an ensemble with 10-100 members is acceptable to keep computations feasible. The data assimilation system in LOTOS-EUROS has been successfully applied for assimilation of O$_3$, NO$_2$, SO$_2$ and aerosol observations from either ground-based or satellite instruments (Barbu et al., 2009; Curier et al., 2012; Eskes et al., 2013; Fu et al., 2017; Schaap et al., 2017; Segers et al., 2010; Timmermans et al., 2009).

### 2.5.2    Assimilation parameter settings

The ensemble specification has a number of settings that need to be set prior to the assimilation experiments. These include the selection of uncertain model parameters (e.g. emissions), the amplitude of the assumed uncertainties and their temporal correlation. Additionally, the assimilation system requires a number of other parameters to be set that influence the results: ensemble size, localization length scale for different types of observations, and representation error covariance. The latter quantifies the differences caused by the different resolutions of the model and the observations.

In this study, we use the selection of model parameters as used in the operational national and European CAMS forecasts, which are the emissions of the ozone precursors (NOx and non-methane volatile organic carbons (NMVOCs)), deposition



velocity of ozone, and the boundary conditions for ozone, all with an assumed uncertainty of 50% and assumed temporal error correlation of 1 day.

The localisation length scale for the ground-based ozone observations is set to 50 km following Curier et al. (2012). The length scale for assimilation of the synthetic $NO_2$ observations has been set to 0 km.

We base the representation error covariance for the satellite observations on observation minus simulation statistics and spatial correlations present in the observations. We obtain the representation error for the surface ozone observations by analysing the impact of spatial averaging. This depends on the spatial variance in the ozone fields, i.e. higher at coastlines and in mountain areas where we find the highest ozone gradients.

An important parameter in the filter is the ensemble size. In general, the ensemble size should be large enough to represent
the covariance structure imposed by model uncertainties and the model physics. In this application, the covariance structure is rather simple, since we describe all uncertainty by the local sources, and observations are available regularly over the domain. With the chosen configuration, we show that an ensemble size of 12 members is sufficient to obtain stable results.

### 2.5.3    Observation data handling

Within the data assimilation system the synthetic observations are ingested using their estimated uncertainties. The available
averaging kernels from the synthetic observations are applied to the model profiles from the surface to the tropopause to make them comparable to the column observations from satellites.

In case the assimilation system is unable to represent an observation correctly, the assimilation may lead to instability of the system. Such an instability can occur if the model lacks certain physical parameterizations or if the model is unable to represent a measurement resulting in a mismatch of the model and measurement spatial or temporal resolution. To avoid this,
we apply a screening procedure to reject those measurements that cannot be represented correctly by the assimilation system. The screening procedure is taken from Järvinen and Undén (1997). If the square of the difference between observation and filter mean is more than a factor 3 larger than the expected variance of this difference, we reject the observation.

In addition, we filter out all synthetic observations with cloud radiance fraction higher than 50% (to exclude cloudy pixels) and those with surface albedo higher than 0.3 (the high reflectance complicates the $NO_2$ retrieval and for OMI data users are
advised to discard scenes with surface albedo values > 0.3 (Boersma et al., 2011)). All observations within the model grid cells are averaged before assimilation using weights according to overlap between pixel footprint and grid cell.

### 2.5.4    AR configurations

Table 1 provides an overview of the different configurations of the assimilation runs. The runs with assimilation of synthetic surface ozone observations only serve as reference runs to represent current operational system capabilities. The runs with
assimilation of GEO, LEO or GEO and LEO observations combined serve to evaluate the added value of the new satellite instruments under study. We perform these runs for a European, zoom and fire domain as presented in Figure 6. The fire




domain covers part of the Iberian Peninsula, and only included in a run for a short episode (first two weeks of August 2003) with large fires in Portugal.

### 2.6 Evaluation method

The goal of this study is the quantitative assessment of the added value of the S5P and S4 $NO_2$ observations on surface air
quality analyses. We performed statistical analysis of the assimilation runs in comparison to the NR to achieve this goal. During the statistical analysis, we use the following diagnostics:

Mean bias: $$MB(X) = \frac{1}{N}\sum(X - NR)$$

Root mean square error: $$RMSE(X) = \sqrt{\frac{1}{N}\sum(X - NR)^2}$$

Temporal correlation: $$R^2 = \left(\frac{\sum(X-\bar{X})(NR-\overline{NR})}{\sqrt{\sum(X-\bar{X})^2\sum(NR-\overline{NR})^2}}\right)^2$$

Normalised Mean Bias: $$NMB(X) = 100 * \frac{\sum(X-NR)}{\sum NR}$$

Mean Absolute Error: $$MAE(X) = \frac{1}{N}\sum|X - NR|$$

Normalised Mean Absolute Error: $$NMAE(X) = 100 * \frac{\sum|X-NR|}{\sum NR}$$

where $X$ are the modelled values from CR; RR or AR, and $NR$ are the values from the Nature Run, and $N$ is the number of values over which we calculate the mean. The value of $N$ varies between the different plots.

## 3  Results

### 3.1 Evaluation of nature run and control run

In the design of an OSSE, it is important to demonstrate that the NR exhibits the same statistical behaviour as the real atmosphere for aspects relevant to the observing system under study. To evaluate this, we compare the ozone and $NO_2$ concentrations from the NR to observations from the AirBase network. We consider the sites falling into the first five classes
as defined in the objective classification method suggested by Joly and Peuch (2012), as we understand them to be representative for the resolution of the NR model. These first five classes mostly encompass sites classified as rural and urban background in the AirBase database but especially for ozone also a small fraction of the urban traffic stations have been assigned to one of the five lowest classes and are therefore considered representative for larger areas. Figure 7 shows that, for ozone, the NR exhibits similar day-to-day and diurnal variability as the observations. On most days, the model
underestimates the afternoon ozone peak, although on some days, the peak is overestimated. For $NO_2$, an underestimation of observed levels is obvious. This could, among others, be due to an underestimation of the emissions prescribed to the model



and/or an overestimation of the vertical mixing. Additionally, the $NO_2$ observations likely overestimate ambient $NO_2$ concentrations due to contamination from other oxidized nitrogen species such as PAN and $HNO_3$ (Giordano et al., 2015; Steinbacher et al., 2007). Despite the bias, the temporal behaviour of the modelled $NO_2$ concentrations from the NR seems to resemble the temporal behaviour of the observations with lowest values during daytime when dilution is strongest and

chemical life time of $NO_2$ is shortest and higher values during the night (see Figure 8). Furthermore, day-to-day variations are reproduced, e.g. the maxima being higher in the second week of July than in the first week.

Figure 8 shows the diurnal cycle from CR, NR and observations averaged over all selected locations for August 2003. For ozone, all three datasets show similar diurnal variation with lowest values in the early morning and an afternoon peak. The models do show an underestimation of the observed ozone values. This bias is regionally dependent (not shown), with larger

negative values for Southern Germany and Central France, small biases over the Netherlands and even positive biases in the MOCAGE model over some periods in Northern Germany. For $NO_2$, the NR from the MOCAGE model is missing the early morning peak due to the morning rush hour when the boundary layer is still shallow and photolysis is still limited. However, considering the daytime hours where satellite observations will be available, the temporal behaviour is comparable, decreasing towards early afternoon and increasing towards the evening.

Regarding the spatial variability, the NR shows strong gradients around large source areas, i.e. cities (e.g. Paris), ports, shipping lanes and highly populated areas with a lot of traffic and industry (e.g. Benelux and Ruhr area) (see Figure 3 bottom plot and Figure 9). This spatial variance is representative of observations from both ground-based observations and satellite observations. We would like to note that the resolution of the NR and synthetic satellite observations (7 by 7 km) is not representative for locations with large sources and high spatial variability at scales < 7 km, therefore not all ground-based

observed variations can be represented.

From the comparisons with real observations, we conclude that the NR is representative of the variability of actual observations over Europe, albeit with a negative bias for both surface $O_3$ and $NO_2$. For the robustness of the OSSE such a negative bias is acceptable as long as the absolute differences between the CR and NR are comparable in size to absolute differences between state-of-the-art models and real observations. This requirement for sufficient differences between the

NR and the CR is an important constraint to avoid the identical twin problem (Arnold and Dey, 1986). Ideally, the differences between NR and CR should show similar features as the differences between the CR and real observations. In Figure 7 and Figure 8 the $O_3$ and $NO_2$ concentrations from the CR (without any assimilation) are plotted in comparison to the NR concentrations and the real AIRBASE observations, respectively. During daytime the differences between the CR and NR are within the same range of values as the differences between CR and the real observations. In the middle of the day,

around the overpass time of the S5 mission, the differences between the CR and the NR are somewhat smaller than between the CR and the real observations, especially for $NO_2$.

For $NO_2$ we assimilate satellite columns, therefore the requirement for avoiding the identical twin problem should also be checked for the column values. The bias in the summer study period $NO_2$ columns between CR and synthetic $NO_2$ columns varies between -5 $10^{15}$ molecules $cm^{-2}$ and 10 $10^{15}$ molecules $cm^{-2}$ (see Figure 10, left plot). Largest biases are seen over the





main source areas and most regions exhibit a bias around a $2 \ 10^{15}$ molecules cm$^{-2}$. Average bias in daytime is between 1 and $2 \ 10^{15}$ molecules cm$^{-2}$ for both summer and winter study periods (not shown). For the summer study period and zoom domain, the average RMSE is around $6 \ 10^{15}$ molecules cm$^{-2}$ (see Figure 12, top left plot, black line), for the entire European domain somewhat lower and for the winter study period slightly higher (not shown). These values are comparable to values

found in Curier et al. (2014) where LOTOS-EUROS model results have been compared to OMI tropospheric NO$_2$ columns over a 6-year period from 2005 to 2010. In that study, the average biases for different domains and periods is 1-2 $10^{15}$ molecules cm$^{-2}$. The RMSE distribution over Europe averaged over the entire 6-year period varies between 0 to 10 $10^{15}$ molecules cm$^{-2}$ similar to our CR differences with the synthetic observations. The bias between CR and synthetic NO$_2$ columns is also comparable to biases of 0.3 to 2 $10^{15}$ molecules cm$^{-2}$ found between the regional CAMS model ensemble and

Multi-Axis Differential Optical Absorption Spectroscopy (MAX-DOAS) surface observations of NO$_2$ columns (Blechschmidt et al., 2017).

Based on above analyses on the realism of the NR and on the differences between CR and NR, we conclude that the main requirements to ensure a robust and realistic OSSE are fulfilled.

## 3.2 Evaluation of OSSE results

### 3.2.1 NO$_2$ columns

Figure 10 shows the bias around time of overpass of S5P (over central Europe 14:00 UTC) between modelled NO$_2$ columns and the synthetic NO$_2$ observations from the CR without any assimilation, and the ARs with assimilation of either S4 or S5P data averaged over the summer period June-July-August (JJA) 2003. The CR overestimates the NO$_2$ columns, especially over the Benelux, Ruhr area in Germany and South-eastern part of the UK. These regions are characterised by high NO$_2$

concentrations due to high density of sources such as traffic, shipping, and industry. A strong decrease of the bias is visible after assimilation of the synthetic Sentinel data in combination with ground-based ozone observations. On average, the impact of the S4 and S5P observations in combination with the ground-based observations is very similar at the overpass time. Note that in the Eastern part of the domain, the bias is not zero for the S5P case but observations do not cover this region around 14:00 UTC. Figure 11 shows that the temporal correlation improves by the assimilation of the Sentinel

observations. Over large parts of the domain and especially the areas with high NO$_2$ concentrations, the temporal correlation increases significantly. Considering the overpass time, the temporal correlation in this case is a measure of the representation of the day-to-day variability. This result shows that the system is working as expected and the data assimilation pulls the modelled values towards the observations.

For the winter period, the conclusion is the same as for the summer period; bias and RMSE decrease while the temporal

correlation increases (not shown).



To evaluate the added value on top of the ground-based observations, we compare the results with the reference run (RR), where we assimilate only ground-based observations. Figure 12 (upper plots) shows the statistical evaluation for the high resolution zoom domain as function of the time of the day, for the modelled $NO_2$ column. We see that for the modelled $NO_2$ columns, the assimilation of ground-based ozone observations in the RR during the summer study period leads, on average,

to a negative impact on RMSE (an increase) during the night and on correlations throughout the entire day. Further investigation demonstrates that the assimilation of the ozone observations improves the surface ozone concentrations. The reason for this negative impact on $NO_2$ column may lie in the fact that the errors in modelled surface ozone concentrations and in modelled $NO_2$ columns are not dominated by the same error source. Another reason could be that the assimilation system is not adapting the dominant parameter responsible for both the ozone and $NO_2$ column errors. Remember there are

four uncertain model parameters in the set-up used in this study that can be adapted through the assimilation. Assimilation of observations should, therefore, always be evaluated with care and one should analyse individual impacts of observation sets or impacts on different components.

The additional assimilation of satellite $NO_2$ column observations improves the RMSE and correlations in $NO_2$ columns in comparison to the RR. This impact is clearer in the results focusing on the fire episode over the Iberian Peninsula (Figure 12,

bottom plots). The impact of the GEO S4 observations is visible throughout the entire day, while the impact of the LEO observations at the overpass time is smaller than the impact of the S4, which continuously feeds the model, and lasts for several hours. The combined assimilation of S4 and S5P observations for the zoom domain only, increases minimally the correlation around the overpass time of S5P in comparison to the assimilation of S4 data only, and does not show a benefit on the RMSE.

**3.2.2**   $NO_2$ surface concentrations

While the results above demonstrate that the system is working as expected and is able to decrease the difference between the model and synthetic $NO_2$ observations, the goal of our study is to investigate the added value of the Sentinel observations for air quality analyses at the surface where the impact of air pollution is the most significant for human health and the ecosystem.

Figure 13 illustrates the impact of the assimilation on modelled surface $NO_2$ concentrations. Here we also see a positive impact of assimilating S4 and/or S5P observations. The RMSE decreases by about 30% during daytime hours (or from the overpass time onwards for S5P) and the correlation increases by about the same amount. On average, this impact is slightly smaller than on the modelled $NO_2$ column. Whereas concentrations of $NO_2$ show very distinct high-resolution features in a small surface layer, mixing towards larger upper layers leads to less pronounced features in the column values. The observed

$NO_2$ columns, therefore, do not contain the level of detail that the surface $NO_2$ does. The comparison with modelled $NO_2$ columns is more direct, covers the same altitude region and therefore, as expected, shows a larger impact than the comparison with surface $NO_2$. As the Sentinel tropospheric $NO_2$ columns do not contain any information on the vertical




distribution within the troposphere, the assimilation of $NO_2$ columns may in some cases also lead to a deterioration of modelled surface concentrations when the vertical $NO_2$ profiles in the model are incorrect.

Figure 14 shows the positive impact on modelled surface $NO_2$ concentrations in a geographical form. This figure shows the bias of the different runs, averaged over the summer period at 10:00 UTC (shortly after the morning rush hour) for a region

centred over France where traffic highways are highly pronounced. The CR overestimates the surface $NO_2$ concentrations over traffic highways and large urban areas (e.g. Paris and London). While the ground-based ozone observations in the RR are able to decrease this positive bias, (averaged over the zoom domain from 0.8 ppb to 0.4 ppb), the S4 observations (available throughout the day) are able to decrease further this bias over the traffic highways (average bias over zoom domain is eliminated). Furthermore, over the shipping lane through the canal between the UK and France, the bias

significantly decreases from values around 5 ppb to nearly 0 ppb. The S5P, with its afternoon overpass time, has an expected negligible impact at 10:00 UTC. As could be seen in Figure 13, at 14:00 UTC the impact of the S5P for surface $NO_2$ concentrations is similar to the impact of S4.

For the winter period, the impact of the assimilation of Sentinel data on surface $NO_2$ concentrations shows a mixed picture, which hints to the importance of the model profile shape for improvements at the surface when assimilating column

observations. A mismatch between the bias in column $NO_2$ and surface $NO_2$ can lead to a negative impact of the satellite observations. Figure 15 shows the bias compared to the NR surface $NO_2$ concentrations for the CR without any assimilation and the AR including the assimilation of surface ozone observations and S4 synthetic observations. The CR shows an overestimation of the NR concentrations over central Europe, large cities in Southern Europe and the shipping lanes in the Mediterranean. Differently, the CR shows an underestimation of surface $NO_2$ concentrations over the North-eastern part of

the domain. The assimilation of the synthetic observations clearly decreases the modelled concentrations over the central European area and leads to a reduced bias over, for example, Germany. Over some of the Eastern European countries, the concentrations increase and the negative bias decreases, for example in Romania, Ukraine and Belarus. However, these positive impacts are not present over the entire domain. Over some areas, the positive bias increases (e.g. over Austria, Slovenia and Northern-Italy) or changes to a negative bias (e.g. over Barcelona, Belgium and the Netherlands). We have

found that, in many of these situations, the bias in surface $NO_2$ concentration does not match the bias in tropospheric $NO_2$ column. For example, in the area covering Austria and Slovenia, the CR underestimates the $NO_2$ column from the NR. Assimilation of synthetic S4 observations derived from the NR then increases the $NO_2$ values and can only do so by increasing the sources at the surface. Even if we would be able to increase only the concentrations at higher altitudes, the satellite measurements do not provide information on the vertical profile or at which altitude the model is biased. The

increased emissions at the surface then lead to even higher concentrations and an increased positive bias in this specific situation. These results demonstrate the importance of a correct representation of the vertical distribution in the model, and of an evaluation of model profiles with independent profile information, for example from MAX-DOAS or aircraft measurements. In our experiment, the vertical distribution of the NR and CR in some cases differs in this way, leading to occasional negative impacts of the assimilation of the synthetic observations.





### 3.2.3    Emissions

We perform the OSSE in this study with an ensemble Kalman filter approach, which optimises the NO₂ concentrations by adjusting the NOx and NMVOC emissions. Figure 16 shows average NOx emission adjustments for the summer period for the RR and AR with S4 data. The distribution of emission adjustments is quite similar between both runs which gives confidence in the choice of uncertain parameters. The assimilation of high-resolution S4 NO2 observations has a relatively large additional impact over some areas of the domain. For example, the NOx emissions over the centre of the Netherlands increase, while the NOx emissions over the shipping route in the English Channel decrease. Due to the high resolution of the S4 data and the smaller length scale set in the assimilation, the S4 observations bring more detailed emission updates than ground-based ozone observations only. Note that this assimilation set-up with uncertainties on the emissions is less flexible at remote locations with small emission fluxes.

## 4    Conclusion

In comparison to preceding instruments (e.g. OMI), the NO₂ observations from S4 and S5P bring considerable advances. These include, 1) much improved resolution, from about 20 km to 7 km or even higher; 2) hourly observations in the case of S4, providing full daytime sampling; and 3) foreseen improvement in the instrument (for TROPOMI an improvement of the slant column uncertainty of 30-40% compared to OMI has been reported (van Geffen et al., 2018) and improvements due to advances in the characterisation of aspects like clouds, albedo and aerosol effects.

In this study, we perform an OSSE experiment to illustrate the added value of these new (S5P) and future (S4) observations for NO₂ analyses over Europe. The OSSE experiment has been carefully designed, with separate models for the NR and AR, and with conservative estimates of the total observation uncertainties. The results show that both S4 and S5P tropospheric NO₂ columns have a clear positive impact on modelled NO₂ values. Assimilation of these observations on top of ground-based ozone observations decreases biases and the RMSE, and improves the temporal variability of modelled NO₂ distributions. S4 will bring a major step forward with its hourly temporal resolution. Owing to the assimilation of synthetic S4 observations, we are able to reconstruct many of the NR features, and the benefit is present throughout the entire day due to the availability of observations at an hourly resolution. For S5P, we observe a good impact up to 3-6 hours after the overpass time. Based on our results, a similar impact is expected for S5 which will have similar technical specifications but an earlier overpass time in the morning. Simultaneous availability of S5 and S5P observations in the future is expected to provide benefits throughout the entire day due to the different overpass times and benefits lasting several hours.

The added value of the satellite observations is visible in both modelled columns as well as in the surface concentrations of NO₂. During the summer period over the zoom domain (Iberian Peninsula), the RMSE in surface NO₂ decreases by about 30% during daytime, while the temporal correlation increases by the same amount. The impact of both instruments on NO₂



columns is even larger. In the winter period, the additional assimilation of the satellite $NO_2$ observations counteracts the positive impact of surface ozone observations in some regions. This results from opposing effects from the bias in satellite $NO_2$ columns and the bias in surface $NO_2$ concentrations, due to different vertical $NO_2$ profiles in the MOCAGE NR and LOTOS-EUROS. It is thus crucial to analyse the model performance for simulating $NO_2$ profiles. Accurate vertical

distributions in the model are a prerequisite for consistent positive results of the column data assimilation at the surface, provided there is no additional information on the vertical profile from the satellite observations.

This study focuses on $NO_2$ analyses, which is one of the main air quality applications that make use of satellite tropospheric $NO_2$ columns. Another common application that combines information from models with satellite observations is the

derivation of emissions. The active data assimilation system based on the ensemble Kalman Filter approach that is applied in this study, is especially suitable when looking at applications such as emission inversions and air quality forecasts, as it does not only update the state of the atmosphere (e.g. $NO_2$ concentrations) but also the driving input parameters (in this study the NOx and NMVOC emissions). A more detailed analysis of the impact of the Sentinel observations on the emissions would be worthwhile to assess the added value of the new $NO_2$ column observations from S4, S5 and S5P for emission inversion

applications.

In October 2017, after completion of our study, S5P has been launched and actual tropospheric $NO_2$ columns have become available. These actual results have proved that our retrieval error estimates, as detailed in Sect. 2.2.4, are conservative due to improvements in the retrievals (van Geffen et al., 2018). For $NO_2$, we find slant column errors for S5P to be of the order

of 0.5-0.6 $10^{15}$ molecules $cm^{-2}$, compared to the 0.7 $10^{15}$ molecules $cm^{-2}$ used in this study. We assume the AMF errors, which dominate the total uncertainty for $NO_2$, and are computed from the cloud and surface albedo uncertainties are comparable to what we use in this study. This means that the weight given to the observations in the data assimilation will be larger with the real observations than with our synthetic observations. The calculated S5P impact on modelled analyses is therefore expected to be on the conservative side. With the arrival of the actual S5P observations we plan to compare results

from assimilation of TROPOMI (S5P) $NO_2$ columns with the results in this study. This comparison will allow evaluation of the realism of the OSSE and will provide valuable support for any future OSSE studies.

This work was part of a study funded by ESA called "Impact of Spaceborne Observations on Tropospheric Composition Analysis and Forecast" (ISOTROP), to study the impact of S4, S5 and S5P observations of ozone, CO, $NO_2$ and HCHO on

air quality analyses. The impact of assimilation of CO from these instruments is presented in Abida et al. (2017). A paper on the impact of assimilation of tropospheric ozone from the instruments is under review (Quesada-Ruiz et al., 2019).





**Code and data availability**

The LOTOS-EUROS CTM is available as open source version for public use via www.lotos-euros.tno.nl. The LOTOS-EUROS data assimilation code used in this study is property of TNO and not allowed to be shared publicly. The MOCAGE model is property of Météo-France and not allowed to be shared publicly.

The volume of the model and synthetic observation datasets discussed in this paper is large, but for scientific purposes subsets can be made available upon request.

**Author contributions**

Renske Timmermans prepared the manuscript with contributions from all authors.

Renske Timmermans, Martijn Schaap, Lyana Curier, Henk Eskes, William Lahoz and Jean-Luc Attié designed the
experiment and provided scientific guidance during the project.

Arjo Segers and Lyana Curier developed the LOTOS-EUROS code and performed the OSSE assimilation runs.

Rachid Abida, Jean-Luc Attié Laaziz El Amraoui, Samuel Quesada and Philippe Ricaud developed the MOCAGE model code, performed the MOCAGE nature run and produced the synthetic ground based ozone observations.

Johan de Haan, Pepijn Veefkind, Henk Eskes, Jukka Kujanpää and Albert Oude Nijhuis produced and provided the synthetic
$NO_2$ observations.

Renske Timmermans, Martijn Schaap, Arjo Segers and Henk Eskes performed the analyses of the model assimilation runs.

*Acknowledgments*

This work was partly supported by the ESA funded project "Impact of Spaceborne Observations on Tropospheric
Composition Analyses and Forecast" (ISOTROP-ESA contract number 4000105743/11/NL/AF)





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





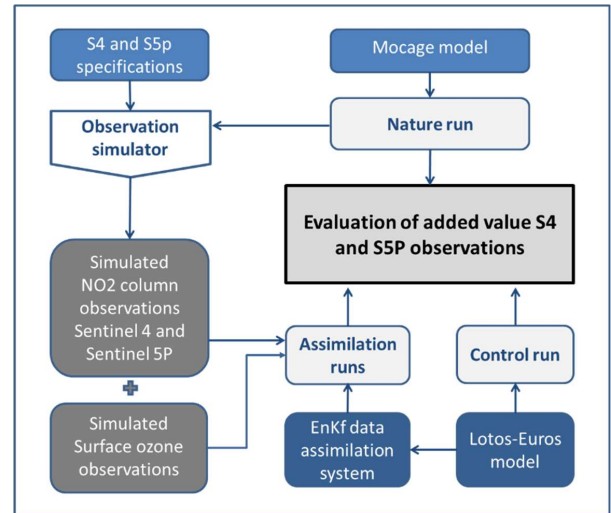

**Figure 1 Diagram of the Observing System Simulation Experiment components following *(Timmermans et al., 2015)*.**

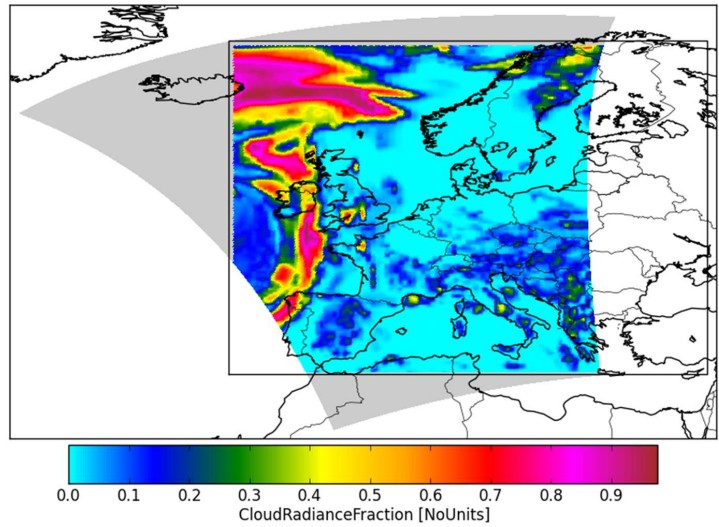

5    **Figure 2 Example of the distribution of cloud radiance fractions for the simulated S5P 12:34 UTC overpass on 1 June 2003. The rectangle presents the European modelling domain.**





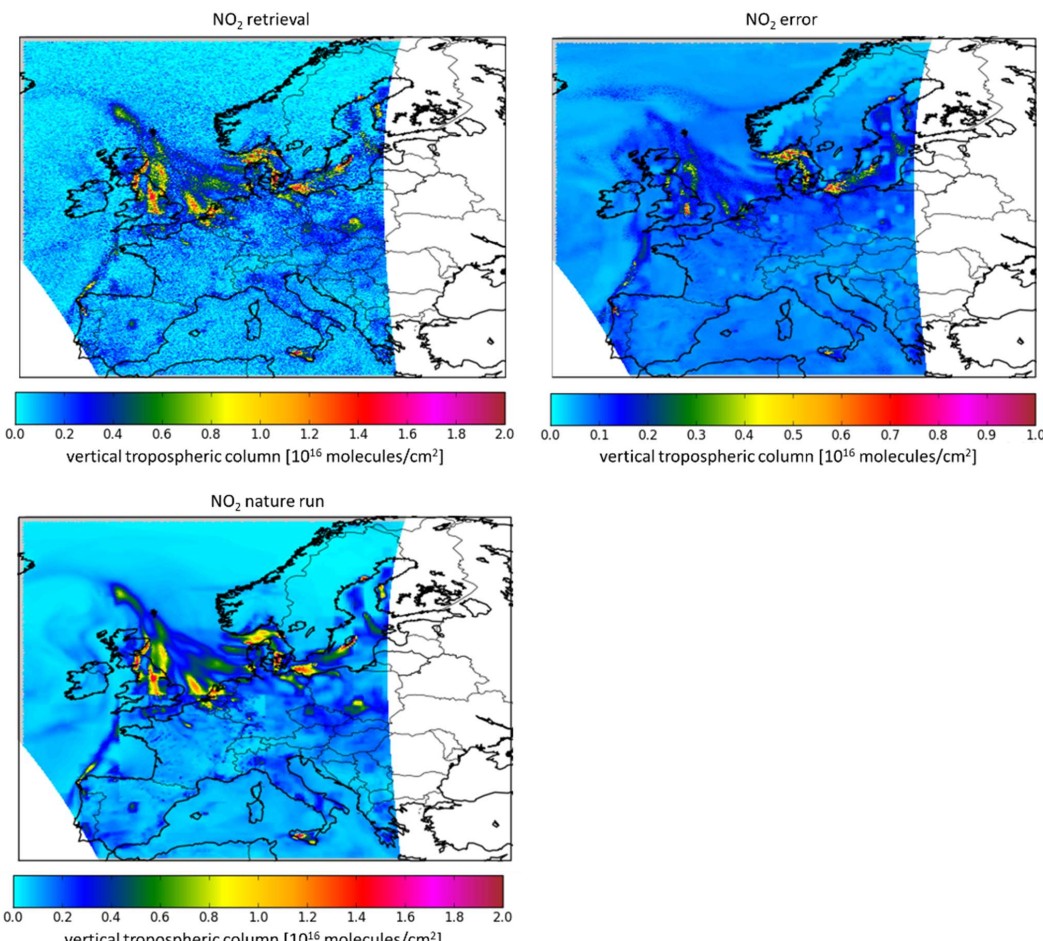

**Figure 3 Example of the S5P synthetic retrieval of the NO₂ tropospheric column on 1 June 2003. The plots show the retrieved, noisy, NO₂ column (top left panel), the associated retrieval error (top right panel) and the NR result (bottom panel).**



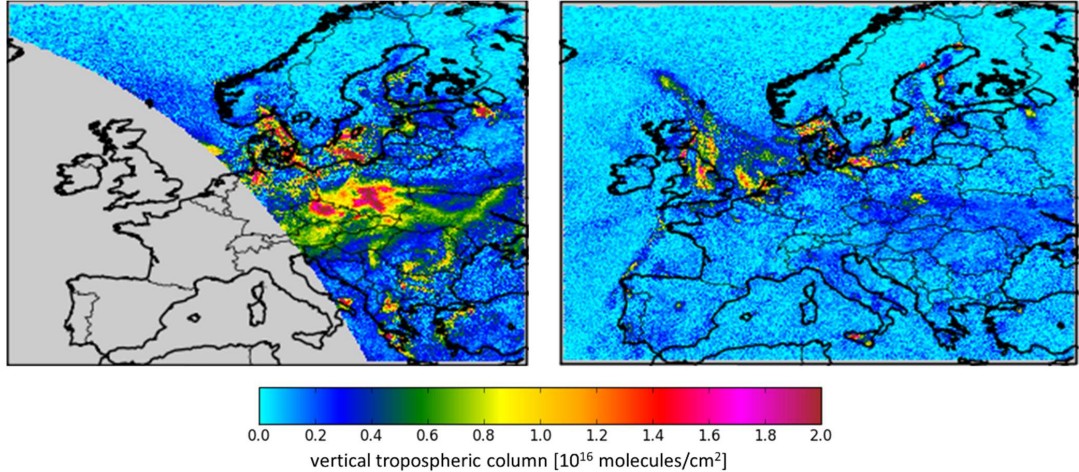

**Figure 4 Example of two S4 NO₂ scenes, early morning, 4:00 UTC, and mid-day, 12:00 UTC, on 1 June 2003. Data is plotted for solar zenith angles < 85 degrees.**

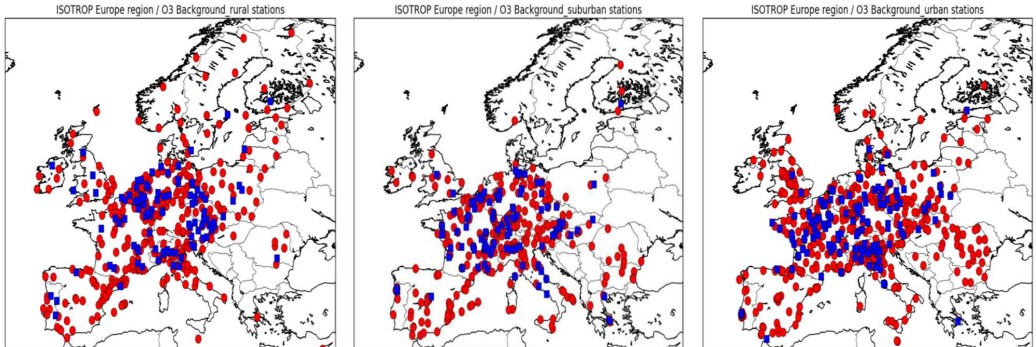

5  **Figure 5 Map of the sampling location for the ozone stations for the European; red circles denote assimilation stations, blue squares denote validation stations. Left plot: rural background stations; Middle plot: suburban background stations; Right plot: urban background stations.**





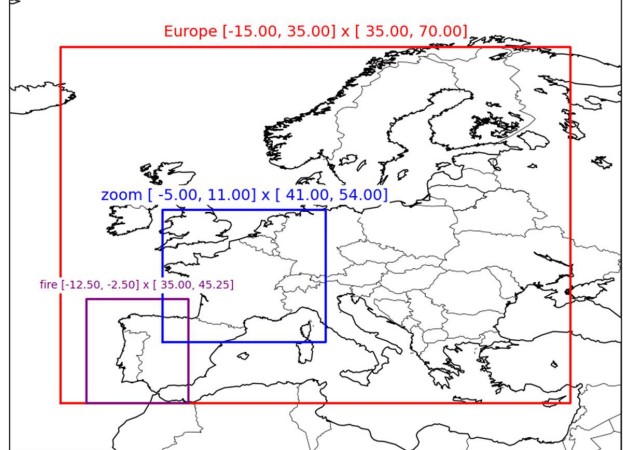

**Figure 6 Illustration of different domains used for the assimilation runs. European domain (red rectangle), zoom domain (blue rectangle) and fire domain (purple rectangle).**

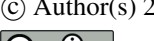


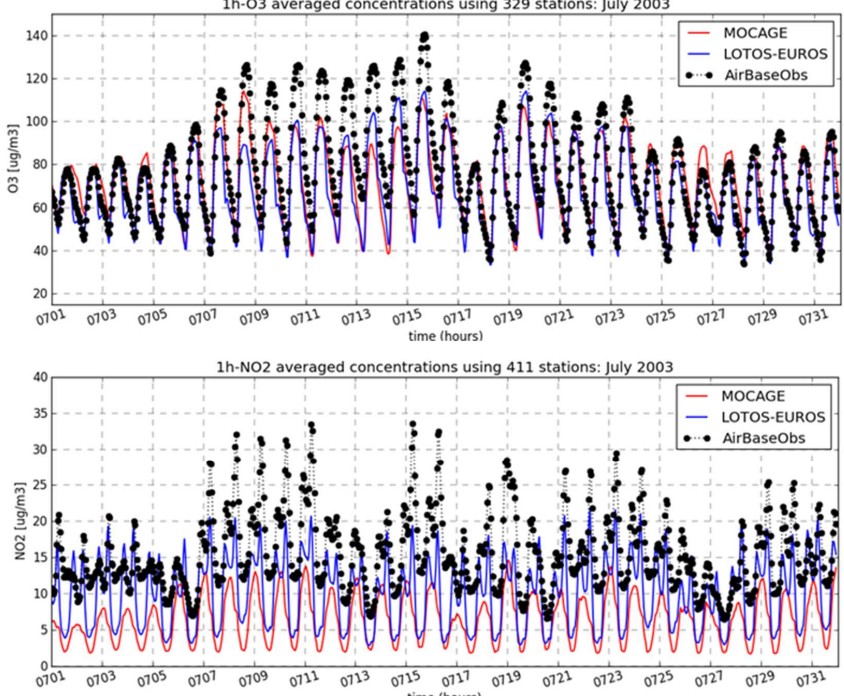

**Figure 7 Time series of ozone (top) and NO₂ (bottom) concentration from NR (red, MOCAGE), CR (blue, LOTOS-EUROS) and AirBase background observations (black dots) for the period of July 2003.**





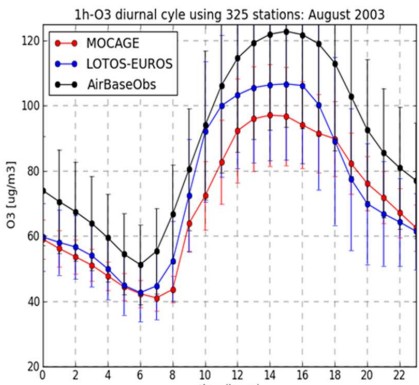 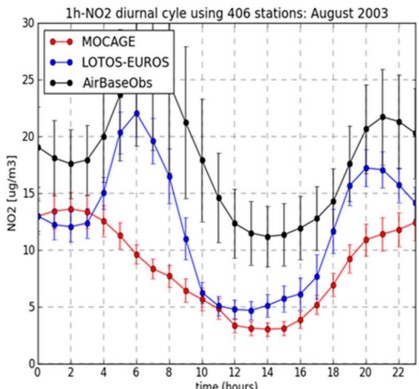

**Figure 8 Diurnal cycle for ozone (left) and NO₂ (right) concentrations from NR (blue, MOCAGE), CR (red, LOTOS-EUROS) and AirBase background observations (black dots) for the period of August 2003. The vertical bars represent the standard deviation over the dataset (different days and stations).**

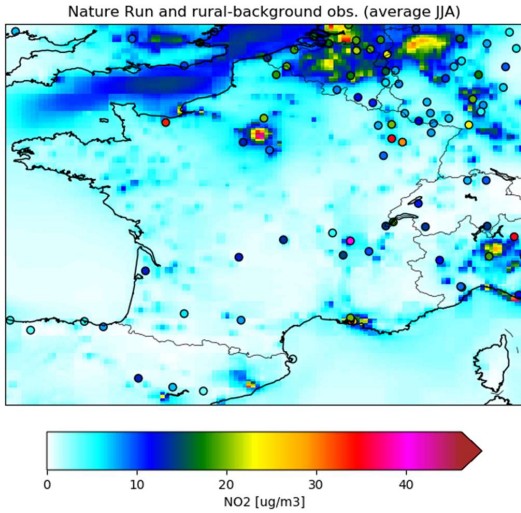

**Figure 9 Average NO₂ concentrations over zoom domain for NR (background) for summer study period. Coloured dots indicate averaged real observation values.**





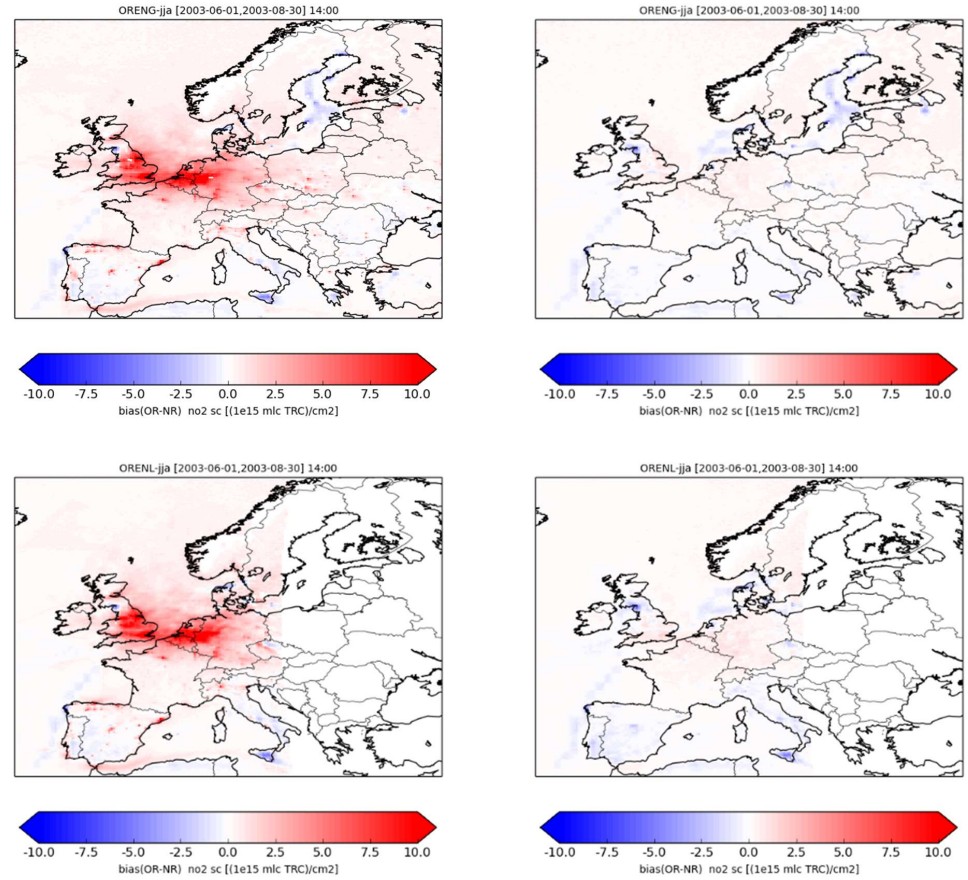

**Figure 10 Summer period (JJA 2003) average NO₂ column bias at 14:00 UTC with synthetic observations for control run without assimilation (left column) and with assimilation of ground-based O₃ +S4 NO₂ (top row) and ground-based O₃ +S5P NO₂ (bottom row).**




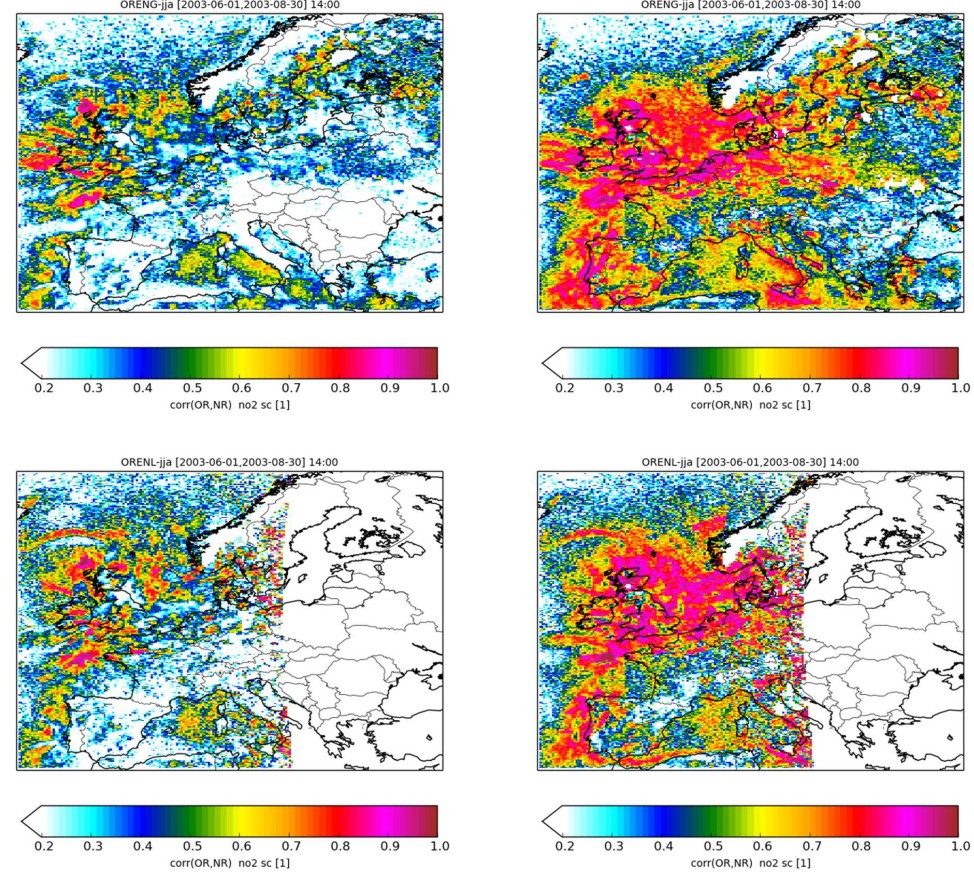

**Figure 11 Summer period (JJA 2003) temporal correlation of modelled NO₂ columns at 14:00 UTC  with synthetic observations for control run without assimilation (left column) and with assimilation of ground-based O₃ +S4 NO₂ (top row) and ground-based O₃ +S5P NO₂ (bottom row).**



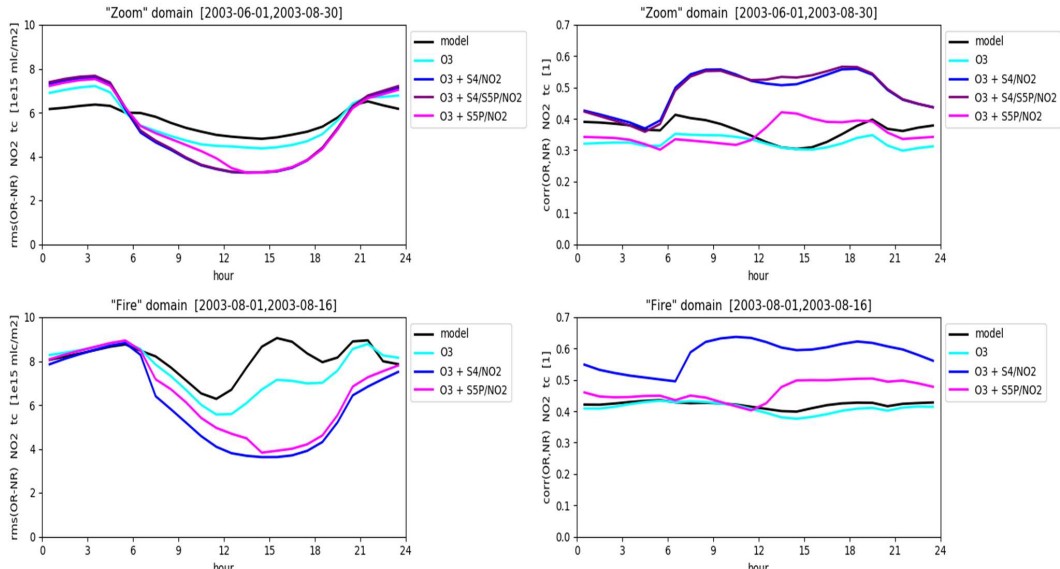

**Figure 12** RMSE (left) and correlation (right) with NR tropospheric NO₂ column for the summer period and zoom domain (top plots) and the fire episode (1-16 August 2013) and domain (bottom plots). Prior (black line) and after assimilation of observations (coloured lines, ground-based O₃ only (cyan), ground-based O₃ + S4 NO₂ (blue), ground-based O₃ + S5P NO₂ (pink) or ground-based O₃ + S4 and S5P NO₂ (purple)).

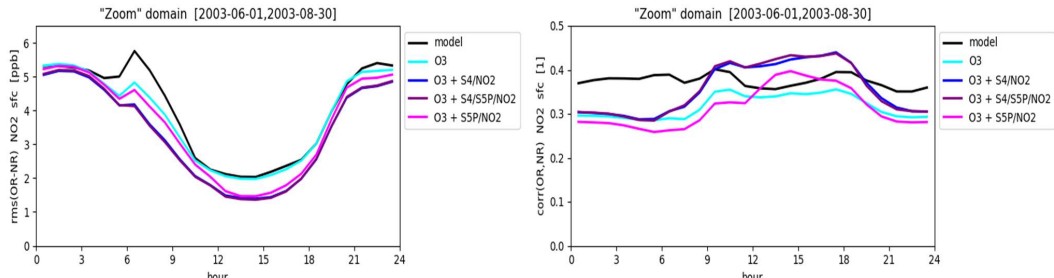

**Figure 13** RMSE (left) and correlation (right) with NR surface NO₂ concentrations for the summer period. Prior (black line) and after assimilation of observations (coloured lines, ground-based O₃ only (cyan), ground-based O₃ + S4 NO₂ (blue), ground-based O₃ + S5P NO₂ (pink) or ground-based O₃ + S4 and S5P NO₂ (purple)).





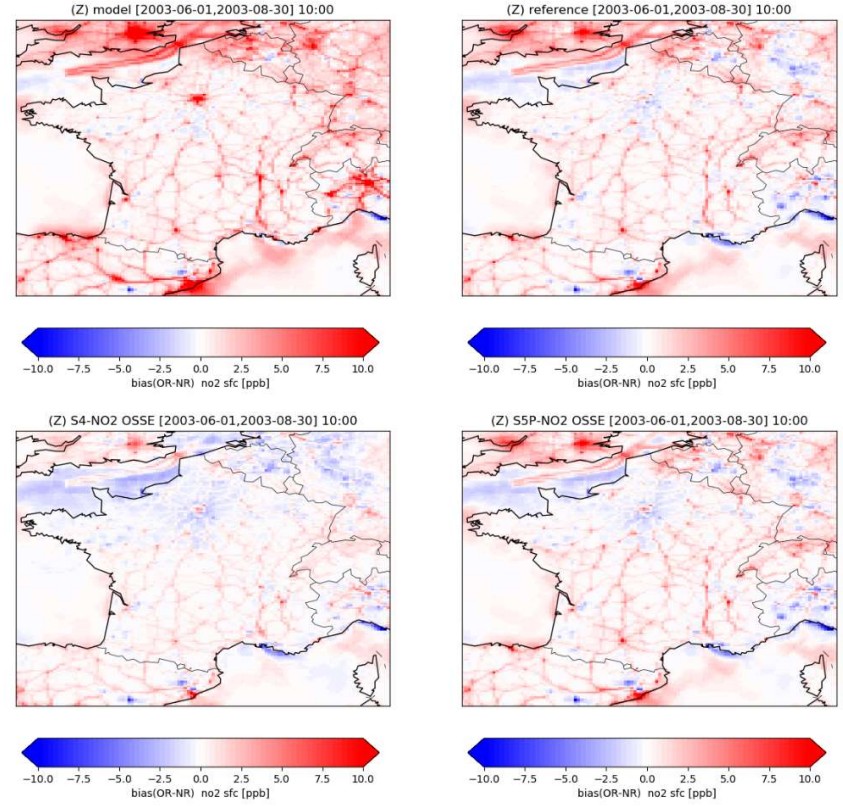

**Figure 14 Zoom domain – summer, 10 UTC average. Bias with NR surface NO₂ before assimilation (top left), and after assimilation of ground-based O₃ (top right), ground-based O₃+S4 NO₂ (bottom left) or ground-based O₃ +S5P NO₂.**




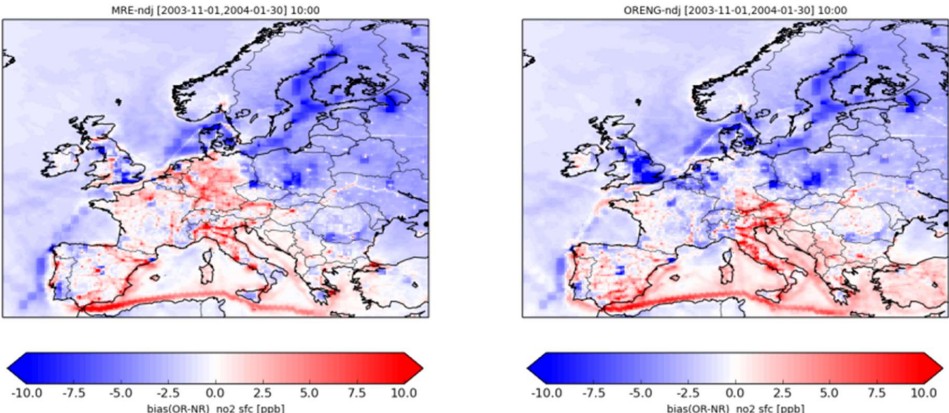

**Figure 15 Winter period average surface NO₂ bias at 10:00 UTC with respect to the NR (in ppb), for the CR without assimilation (left plot) and with assimilation of ground-based O₃ +S4 NO₂ (right plot).**

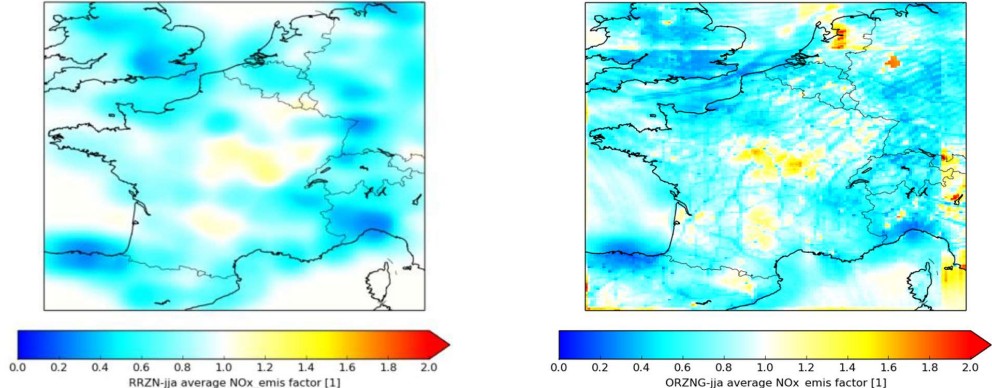

5 **Figure 16 Average NOx emission increment for the run with assimilation of ground-based ozone observations only (RR) (left) and the S4 assimilation run (right) for the summer study period over the zoom domain.**





Table 1 Overview of assimilation run configurations. RR: Reference Run; OR: Assimilation Run; F: fire domain; Z: Zoom domain; E: European domain; GN: GEO satellite; LN: LEO satellite; LGN: LEO and GEO satellite.

| Run ID | Run | Domain | Resolution (degrees) | Assimilation | |
|--------|-----|--------|----------------------|--------------|--------------|
| | | | | Ground | Satellite |
| RRF | Reference Run | fire | 0.0625x0.125 | Surface ozone | No |
| RRZ | | zoom | 0.0625x0.125 | Surface ozone | No |
| RRE | | Europe | 0.125x0.25 | Surface ozone | No |
| ORFGN | AR GEO | fire | 0.0625x0.125 | Surface ozone | GEO/S4 NO2 |
| ORZGN | | zoom | 0.0625x0.125 | Surface ozone | GEO/S4 NO2 |
| OREGN | | Europe | 0.125x0.25 | Surface ozone | GEO/S4 NO2 |
| ORFLN | AR LEO | fire | 0.0625x0.125 | Surface ozone | LEO/S5P NO2 |
| ORZLN | | zoom | 0.0625x0.125 | Surface ozone | LEO/S5P NO2 |
| ORELN | | Europe | 0.125x0.25 | Surface ozone | LEO/S5P NO2 |
| ORZLGN | AR GEO+LEO | zoom | 0.0625x0.125 | Surface ozone | GEO/S4 NO2 LEO/S5P NO2 |