# Peer review of "Impact of synthetic spaceborne NO2 observations from the Sentinel-4 and Sentinel-5p platforms on tropospheric NO2 analyses"

_Atmospheric Chemistry and Physics, 2018_

## Referee Comment (RC1) · Anonymous Referee #1 · 8 Apr 2019

The submitted manuscript deals with the very timely issue of making the best possible use of new upcoming Sentinel satellite products related to air quality. I found the manuscript very clear, concise and informative. It is basically ready for publication on ACP, I only have a very few requests for clarification/improvement:

- Regarding the estimation of ground-based observational uncertainties, I wonder if this work could be useful: Thunis et al. "Model quality objectives based on measurement uncertainty. Part I: Ozone", Atm. Env. 79, November 2013, Pages 861-868.

- Models description in sec. 2.1 and 2.4: it would be useful to add information on the method/source of emission splitting/speciation, which is currently missing. E.g. how

[Figure]

NOx emissions are split in NO/NO2 or other nitrogen compounds? How the total VOCs emissions are speciated and then lumped in the model specific chemical mechanism? Also the information on chemical mechanisms used would be a useful to be added.

- On the EnKF corrections: in sec. 2.5.2 are briefly explained the "parameters" that are optimized by the assimilation procedure. However, in the rest of the manuscript, only the correction to emissions is shortly discussed (Fig. 16 and related text). What about the other parameters, e.g. ozone deposition velocity and boundary conditions (or others)? A brief mention to these would be also informative.

- Fig. 13: I suggest to revise the lines in the figure for improved readability. Indeed, on print the black, blue and purple look quite similar. Perhaps adding also a variation to the line style (e.g. dashed, dotted, etc.) would help.
* * *

---

## Referee Comment (RC2) · Anonymous Referee #2 · 25 Jun 2019

**Referees Report**

In general the paper is scientifically sound in approach and methodology and addresses subject matter of general interest in light of the new or planned satellite missions targeting atmospheric composition and air quality.

Note that the information provided about Sentinel-4 is still not at all clear or accurate in some aspects – it is referred to a satellite at one point when it is in-fact an instrument that will fly on the MTG-S platform. These issues need correction. Some recommendations are made below in the detailed comments.

**Detailed Comments**

1) Page 2 Line 28: The nominal spatial resolution of GOME-2 is 40 (ALT) x 80 (ACT) km
2) Page 3 Line 5: information of -> information on
3) Page 3 Line 11: analyses -> analysis
4) Page 3 Line 18: Replace "will be followed up by the Sentinel-5 (S5) mission planned for launch in 2021." -> will be followed by the Sentinel-5 (S5) mission to be flown on the EUMETSAT EPS-SG A satellite, planned for launch in 2022 (https://www.eumetsat.int/website/home/Satellites/FutureSatellites/EUMETSATPolarSystemSecondGeneration/index.html)
5) Page 3 Line 21: insight on -> insight into
6) Page 3 Line 22 & 23: "The satellite flies in an early afternoon sun-synchronous orbit with an equator crossing mean local solar time of 13:30 with a wide swath enabling" – the satellite does not have a wide swath, the instrument does. Please rephrase.
7) Page 3 Lines 26-19: "The Ultra-violet/Visible/Near-Infrared (UVN) sounder as part of the Sentinel-4 (S4) mission on board the geostationary (GEO) Meteosat Third Generation Sounder (MTG-S) satellite (ESA, 2018a), with a planned launch in 2021, will also provide similar resolution to TROPOMI, but with higher temporal resolution (hourly)." Should be replaced with "The Sentinel-4 mission (ESA, 2018a) is implemented as the Ultra-violet/Visible/Near-Infrared (UVN) sounder to be flown on the Meteosat Third Generation Sounder (MTG-S) satellite (https://www.eumetsat.int/website/home/Satellites/FutureSatellites/MeteosatThirdGeneration/index.html) with a planned launch in 2023. It will provide similar resolution to TROPOMI, but with higher hourly temporal resolution."
8) Page 3 Line 14: overoptimistic -> overly optimistic
9) Page 4 Line 27: influencing parameters for the observations -> parameters which influence the observations
10) Page 5 Line 3: overoptimistic -> overly optimistic
11) Page 5 Line 8: set-up -> set up
12) Page 5 Line 19: in accordance to -> in accordance with
13) Page 6 Line 6: Sentinel-4 is not a satellite (see comment above). Please correct to be on-board the MTG-S satellite.
14) Page 6 Line 8: wavelengths ranges -> wavelength ranges
15) Page 6 Line 15: S4 orbit -> MTG-S orbit
16) Page 6 Line 25: Sentinel-4 is not a satellite (see comment above). Please correct to be on-board the MTG-S satellite.
17) Page 6 Line 27: S4 orbit -> MTG-S orbit
18) Page 6 Lines 29 & 30: Based on the location of the edges, we compute the coordinates of the individual observations assuming a spatial resolution of 7x7 km2 at nadir for both

instruments. This statement applies to S5P but not S4. There is no nadir for Sentinel-4. It is embarked on a geostationary platform (located in the equatorial plane) and the FOV is off nadir. The nominal resolution is 8 x 8 km at a nominal location within Europe. Please correct.

19) Page 7 Line 1: "Note that normally the size of the footprints away from nadir increases roughly like 1/viewing angle." This should be clarified for Sentinel-4.

20) Page 8 Line 12: Please replace the Sentinel-4 and Sentinel-5 Mission Requirements document with https://sentinel.esa.int/documents/247904/2506504/Copernicus-Sentinels-4-and-5-Mission-Requirements-Traceability-Document.pdf

---

## Author Comment (AC1) · 21 Jul 2019

Thank you for your review of our paper. Please find below our responses to the comments in the following structure:

(1) comments from Referees, (2) author's response, (3) author's changes in manuscript.

Anonymous Referee 1 - Regarding the estimation of ground-based observational uncertainties, I wonder if this work could be useful: Thunis et al. "Model quality objectives based on measurement uncertainty. Part I: Ozone", Atm. Env. 79, November 2013,

[Figure]

Pages 861-868.

Thank you, this is a very useful and interesting paper, we have added this reference in the section on observational errors.

- Models description in sec. 2.1 and 2.4: it would be useful to add information on the method/source of emission splitting/speciation, which is currently missing. E.g. how NOx emissions are split in NO/NO2 or other nitrogen compounds? How the total VOCs emissions are speciated and then lumped in the model specific chemical mechanism? Also the information on chemical mechanisms used would be a useful to be added.

We understand that this would be useful information, however we believe adding all this information would unnecessarily lengthen the paper. We have therefore decided to add a little bit more information and refer to relevant model description papers and report for those interested in more details. Added text: The gas-phase chemistry in the LOTOS-EUROS model is based on a modified version of the CBMIV mechanism. We refer to Manders-Groot et al. (2016) and Manders et al. (2017) for more details on the emission speciation and corresponding lumping to the chemical mechanism species. The gas-phase chemistry in MOCAGE uses the RACMOBUS chemical scheme, a combination of the Regional Atmospheric Chemistry Mechanism tropospheric scheme -RACM- (Stockwell et al., 1997) and the REactive Processes Ruling the Ozone BUdget in the Stratosphere stratospheric scheme -REPROBUS- (Lefèvre et al., 1994).

- On the EnKF corrections: in sec. 2.5.2 are briefly explained the "parameters" that are optimized by the assimilation procedure. However, in the rest of the manuscript, only the correction to emissions is shortly discussed (Fig. 16 and related text). What about the other parameters, e.g. ozone deposition velocity and boundary conditions (or others)? A brief mention to these would be also informative.

Indeed the assimilation system also optimizes ozone deposition velocity and boundary conditions, however the changes in these parameters are more related to ozone and is small compared to the emission changes in NOx, when assimilating NO2. We added a

brief mention of the other parameters in the section where we show the adjusted emissions: ….which optimises the NO2 concentrations by specification of uncertainties in model parameters, as described in section 2.5.2. The parameter that is most directly influencing the NO2 concentrations is the NOx emission; other parameters such as NMVOC emissions and ozone deposition velocities are more related to ozone. As example of the change in model parameters, Figure 16 shows average NOx emission adjustments . . .. . ...

- Fig. 13: I suggest to revise the lines in the figure for improved readability. Indeed, on print the black, blue and purple look quite similar. Perhaps adding also a variation to the line style (e.g. dashed, dotted, etc.) would help.

We have changed the line colors and added dotted and dashed lines for printed versions.

---

## Author Comment (AC2) · 21 Jul 2019

Thank you for reviewing our paper, please find below our response to the comments

Anonymous Referee 2 - Note that the information provided about Sentinel-4 is still not at all clear or accurate in some aspects – it is referred to a satellite at one point when it is in-fact an instrument that will fly on the MTG-S platform. These issues need correction. Some recommendations are made below in the detailed comments.

Thank you for point this out to us, we have corrected the erroneous information, see reaction to detailed comments below.

[Figure]

Detailed Comments 1) Page 2 Line 28: The nominal spatial resolution of GOME-2 is 40 (ALT) x 80 (ACT) km

Corrected.

2) Page 3 Line 5: information of -> information on

Corrected.

3) Page 3 Line 11: analyses -> analysis

Corrected.

4) Page 3 Line 18: Replace "will be followed up by the Sentinel-5 (S5) mission planned for launch in 2021." -> will be followed by the Sentinel-5 (S5) mission to be flown on the EUMETSAT EPS-SG A satellite, planned for launch in 2022 (https://www.eumetsat.int/website/home/Satellites/FutureSatellites/EUMETSATPolarSystemSecondGeneration/index.html

Corrected

5) Page 3 Line 21: insight on -> insight into

Corrected

6) Page 3 Line 22 & 23: "The satellite flies in an early afternoon sun-synchronous orbit with an equator crossing mean local solar time of 13:30 with a wide swath enabling" – the satellite does not have a wide swath, the instrument does. Please rephrase.

Rephrased. The satellite flies in an early afternoon sun-synchronous orbit with an equator crossing mean local solar time of 13:30 with a wide swath instrument enabling daily global coverage but limiting the temporal coverage to one or two daytime observations per day at mid-latitudes

7) Page 3 Lines 26-19: "The Ultra-violet/Visible/Near-Infrared (UVN) sounder as part of the Sentinel-4 (S4) mission on board the geostationary (GEO) Meteosat Third Generation Sounder (MTG-S) satellite (ESA, 2018a), with a planned

launch in 2021, will also provide similar resolution to TROPOMI, but with higher temporal resolution (hourly)." Should be replaced with "The Sentinel-4 mission (ESA, 2018a) is implemented as the Ultra-violet/Visible/Near-Infrared (UVN) sounder to be flown on the Meteosat Third Generation Sounder (MTG-S) satellite (https://www.eumetsat.int/website/home/Satellites/FutureSatellites/MeteosatThirdGeneration/index.html) with a planned launch in 2023. It will provide similar resolution to TROPOMI, but with higher hourly temporal resolution."

Changed

8) Page 3 Line 14: overoptimistic -> overly optimistic

Corrected

9) Page 4 Line 27: influencing parameters for the observations -> parameters which influence the observations

Corrected

10) Page 5 Line 3: overoptimistic -> overly optimistic

Corrected

11) Page 5 Line 8: set-up -> set up

Changed

12) Page 5 Line 19: in accordance to -> in accordance with

Corrected

13) Page 6 Line 6: Sentinel-4 is not a satellite (see comment above). Please correct to be on-board the MTG-S satellite.

We have corrected it to: TROPOMI instrument on board the S5P satellite and the S4/UVN instrument on board the MTG-S satellite.

14) Page 6 Line 8: wavelengths ranges -> wavelength ranges

Corrected

15) Page 6 Line 15: S4 orbit -> MTG-S orbit

Corrected

16) Page 6 Line 25: Sentinel-4 is not a satellite (see comment above). Please correct to be on-board the MTG-S satellite.

We have corrected it to: TROPOMI instrument on board the S5P satellite and the S4/UVN instrument on board the MTG-S satellite.

17) Page 6 Line 27: S4 orbit -> MTG-S orbit

Corrected

18) Page 6 Lines 29 & 30: Based on the location of the edges, we compute the coordinates of the individual observations assuming a spatial resolution of 7x7 km2 at nadir for both instruments. This statement applies to S5P but not S4. There is no nadir for Sentinel-4. It is embarked on a geostationary platform (located in the equatorial plane) and the FOV is off nadir. The nominal resolution is 8 x 8 km at a nominal location within Europe. Please correct.

We have now focused the first part of the paragraph on S5p, to avoid the confusion, see also next comment.

19) Page 7 Line 1: "Note that normally the size of the footprints away from nadir increases roughly like 1/viewing angle." This should be clarified for Sentinel-4. This indeed is valid for Sentinel 5 and S4. We have added the following sentence: For S4 the viewing angle decreases with latitude, and pixels become stretched from roughly 7 km in north-south direction at 40N up to 25 km at 70N.

20) Page 8 Line 12: Please replace the Sentinel-4 and Sentinel-5 Mission Requirements document with https://sentinel.esa.int/documents/247904/2506504/Copernicus-Sentinels-4-and-5-Mission-Requirements-Traceability-Document.pdf

Corrected